# The Economics of Rapid Multiplication of Hybrid Poplar Biomass Varieties

**Brian J. Stanton \*, Kathy Haiby, Carlos Gantz, Jesus Espinoza** 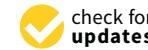 **and Richard A. Shuren**

GreenWood Resources 1500 SW 1st Avenue, Portland, OR 97201, USA; kathy.haiby@gwrglobal.com (K.H.);
carlos.gantz@gwrglobal.com (C.G.); jesus.espinoza@gwrglobal.com (J.E.); rshuren@comcast.net (R.A.S.)
**\*** Correspondence: brian.stanton@gwrglobal.com; Tel.: +1-971-533-7037

**Abstract:** Background: Poplar (*Populus* spp.) hybridization is key to advancing biomass yields and conversion efficiency. Once superior varieties are selected, there is a lag in commercial use while they are multiplied to scale. Objective: The purpose of this study was to assess the influence of gains in biomass yield and quality on investment in rapid propagation techniques that speed the time to commercial deployment. Material and Methods: A factorial experiment of propagation method and hybrid variety was conducted to quantify the scale-up rate of in vitro and greenhouse clonal multiplication. These data were used in modeling the internal rate of return (IRR) on investment into rapid propagation as a function of genetic gains in biomass yield and quality and compared to a base case that assumed the standard method of supplying operational varieties in commercial quantities from nurseries as hardwood cuttings, capable of yields of 16.5 Mg ha$^{-1}$ year$^{-1}$. Results: Analysis of variance in macro-cutting yield showed that propagation method and varietal effects as well as their interaction were highly significant, with hedge propagation exceeding serial propagation in macro-cutting productivity by a factor of nearly 1.8. The *Populus deltoides* × *P. maximowiczii* and the *Populus trichocarpa* × *P. maximowiczii* varieties greatly exceeded the multiplication rate of the *P.* × *generosa* varieties due to their exceptional response to repeated hedging required to initiate multiple tracks of serial propagation. Analyses of investment into rapid propagation to introduce new material into plantation establishment followed by a 20-year rotation of six coppice harvests showed that gains in biomass yield and quality are warranted for a commitment to rapid propagation systems. The base case analysis was generally favored at yields up to 18 Mg$^{-1}$ year$^{-1}$ dependent on pricing. The rapid multiplication analysis proved superior to the base case analysis at the two highest yield levels (27.0 and 31.5 Mg ha$^{-1}$ year$^{-1}$,) at all price levels and at yields of 22.5 Mg$^{-1}$ year$^{-1}$, dependent on price and farm location. Conclusion: Rapid multiplication is a reliable method to move improved plant material directly into operations when valued appropriately in the marketplace.

**Keywords:** hybrid poplar; genetic improvement; clonal propagation; biofuels; renewable energy

---

## 1. Introduction

The profitability of hybrid poplar biomass production and its biochemical conversion to transportation fuels is dependent on genetically improved interspecific varieties. Both farm and refinery economics are influenced by genetic gains in biomass yield [1,2] and biomass quality, the latter manifested in the ease of cell wall deconstruction and sugar release during pretreatment and hydrolysis [3–5]. The improvement cycle is not trivial, encompassing up to seven years to complete the process of hybridization, clonal propagation, and yield testing through the first coppice cycle and laboratory analyses of biomass composition and conversion efficiency. Additionally, once superior varieties have been identified, it remains for them to be multiplied to levels for widespread initial varietal deployment. Considering the density of bioenergy plantations approximates 3500–4500 stems

per hectare (ha) and the sizable acreage that needs to be cultivated to produce the tonnage required by refinery operations, the time to fully expand the supply of planting stock of newly-selected varieties impedes the expeditious delivery of gains in yield and conversion efficiency. While traditional nursery propagation may take seven years and 150 ha to introduce a new varietal into a moderately sized plantation (i.e., 10,000 ha), a combination of laboratory and greenhouse propagation has demonstrated a condensed delivery timeline [6]. An argument for such rapid multiplication propagation techniques as indispensable to the initial propagation of new hybrid varieties for bioenergy farms has been made since the beginnings of clonal forestry [7].

Commercial biomass farms employing interspecific poplar hybrids between sections Aigeiros and Tacamahaca are normally planted as clonal stands using inexpensive one-year-old hardwood cuttings that establish in the field by formation of adventitious roots that elongate from initials formed on nursery stock the previous growing season during shoot development. Although measurable genetic variance in vegetative propagation has been reported, field rooting of Aigeiros × Tacamahaca hybrid taxa using hardwood cuttings is a generally reliable propagation method [8]. However, for the initial scaling of clonal selections, varieties are rapidly multiplied in greenhouses using succulent cuttings rooted under mist propagation in soil or hydroponically to produce containerized planting stock [9,10]. Alternatively, in vitro micropropagation systems of exceedingly greater capacity are available, albeit costly [11,12]. Recent micropropagation research—producing plantlets in liquid-phase bioreactors for ex vitro greenhouse rooting—may ultimately prove cost-effective for industrial clonal propagation [13], but as currently practiced, micropropagation is prohibitively expensive for commercial planting stock quantities [14]. Until the affordability of in vitro systems is proven, the rapid multiplication technique as developed in Finland for hybrid aspen (*Populus* × *wettsteinii* Hämet-Ahti) may be the most cost-effective avenue for scaling new selections [15]. This approach utilizes in vitro micro-cuttings that are produced in laboratories, rooted in greenhouses, and subsequently expanded by hedge propagation of macro-cuttings to generate multiple serial propagation tracks. In vitro propagation takes place in a laboratory using culture media and glass vessels [16]. Hedge propagation is the continuous harvesting of re-sprouting plants for re-planting under greenhouse conditions. Serial propagation begins with greenhouse propagation of cuttings that become primary ramets, from which cuttings are collected that become secondary ramets that lead to tertiary ramets and so on.

The economics of this combined approach has not been studied for the Aigeiros × Tacamahaca taxa, although generally recognized to be more expensive than conventional nursery propagation. A factorial study was therefore conducted to assess the economics of a rapid multiplication method based on the propagation of micro- and macro-cuttings for Aigeiros × Tacamahaca hybrid varieties patterned on the Finnish *P.* × *wettsteinii* model. For this paper, micro-cuttings are shoots produced through multiple rounds of in vitro propagation from explants containing a shoot-tip or nodal meristem. Conversely, shoots produced using ex vitro greenhouse rooting techniques for containerized planting stock are referred to as macro-cuttings in this study. The intent of the investigation was to: (1) Quantify the efficiency of greenhouse hedge and serial macro-cutting multiplication initiated with in vitro micro-cuttings for several varieties of diverse taxa and (2) determine the profitability of greenhouse rapid multiplication as a function of genetic gains in biomass yield and biomass quality using the internal rate of return as a standard metric of economic performance [17].

## 2. Materials and Methods

### 2.1. Selection of Plant Material

Twenty-two centimeter (cm) hardwood cuttings of four experimental varieties were collected in January 2013 from a clonal bank maintained by GreenWood Resources at its Tree Improvement Center, Westport, Oregon, USA. The varieties were chosen to provide a contrast of nursery growth rates and four distinct hardwood cutting production categories. A further consideration in the choice was the assemblage of four distinct taxa, *Populus* × *generosa* (Henry) and its reciprocal and hybrids formed from

separate crosses between *Populus deltoides* (Bartram ex Marsh.) and *Populusmaximowiczii* (Henry) and *Populus balsamifera* subsp. *trichocarpa* (Torr. and Gray) (Table 1). The hardwood cuttings were used to grow a single stock plant for each variety in 7.6 cubic decimeters ($dm^3$) pots in a greenhouse to provide nodal cuttings, with which in vitro propagation was initiated. Before entering dormancy in the fall, the stock plants were sheared during the 2013 growing season to encourage branching. Stock plants were forced in January 2014 and grown for three months to an average height of 92 cm when succulent cuttings were collected from axillary and terminal shoots. The shoots were trimmed to a terminal bud and two nodal buds, disinfected with a fungicidal application, refrigerated, and shipped to a contract micropropagation laboratory for in vitro production of micro-cuttings for greenhouse rooting trials. Between 47 and 87 succulent shoots approximately 10 to 14 cm in length were provided in March–July 2014 for culture establishment and shoot proliferation.

**Table 1.** Experimental varieties, taxa, and hardwood nursery production metrics.

| Varietal Identity | Taxon [1] | Sprouts Stool$^{-1}$ | DBH [2] (mm) | Height [2] (dm) | Cutting Utilization [3] | Production Category |
|---|---|---|---|---|---|---|
| 790-99-28596 | T × D | 3.8 | 17 | 31 | 0.47 | Moderate growth; low usage |
| 846-00-30120 | D × M | 1.7 | 26 | 40 | 0.29 | Excessive growth; low usage |
| 854-00-30517 | T × M | 2.3 | 21 | 42 | 0.72 | Excessive growth; high usage |
| 893-01-31899 | D × T | 1.5 | 18 | 33 | 0.56 | Moderate growth and usage |

[1] Taxa coded as: D × T and T × D (*Populus deltoides* × *P. balsamifera* subsp. *trichocarpa* and reciprocal, aka *Populus* × *generosa*), T × M (*Populus balsamifera* subsp. *trichocarpa* × *P. maximowiczii*), D × M (*Populus deltoides* × *P. maximowiczii*). [2] Breast-height diameter and height of the largest sprout per stool following the fourth coppice year. [3] Proportion of sprout length without sylleptic branching from which 24 cm cuttings bearing axillary buds required for quality hardwood cuttings.

Five hundred micro-cuttings were returned to the greenhouse and transplanted into 144-cell trays (23 $cm^3$ capacity) filled with a commercial soil mix in the spring of 2015 and grown under mist-propagation using artificial lighting to extend daylength to 18 h. Temperatures were set at 23 °C for acclimation and root initiation. The total number of surviving ramets was recorded for each variety after two months, at which time they ranged from five to 10 cm in height. The tallest 50 rooted plants of each variety were then transplanted into plastic containers (163 $cm^3$ capacity) filled with the same soil mix to serve as a hedge bank. These were arranged in racks by variety at a density of 37 $cm^2$ per plant. The plants were grown through 2015, allowed to go dormant, and then forced in January 2016. Succulent 20–22 mm (mm) macro-cuttings were collected in March 2016 from the hedges and used in establishing one track of serial propagation that was cycled through the quinary ramet stage (Figure 1). The 50-ramet hedges were harvested five additional times in propagating additional primary ramets throughout all four seasons.

*2.2. Greenhouse Rooting Trials*

Succulent macro-cuttings were dipped in a commercial rooting hormone (1000 ppm indole acetic acid and 500 ppm naphthalene acetic acid) during hedge and serial propagation before sticking in to 200 cell trays (20 $cm^3$ capacity). Trays were grown under an 18 h day length at 23 °C and misted every 20 min for 10 s for the first week of acclimation. Mist settings were changed to a 30 min frequency of 10 s duration for the second week. Thereafter, the plants were misted every 60 min for 20 s. The schedule of hedge propagation cycles was not coincident with the schedule of the serial propagation due to the time to regrow the hedges following the first harvest to initiate serial propagation (Table 2). Thereafter,

the cycles of hedge propagation lagged the serial propagation cycles by approximately six weeks. Succulent macro-cuttings were harvested after six to eight weeks dependent upon the season.

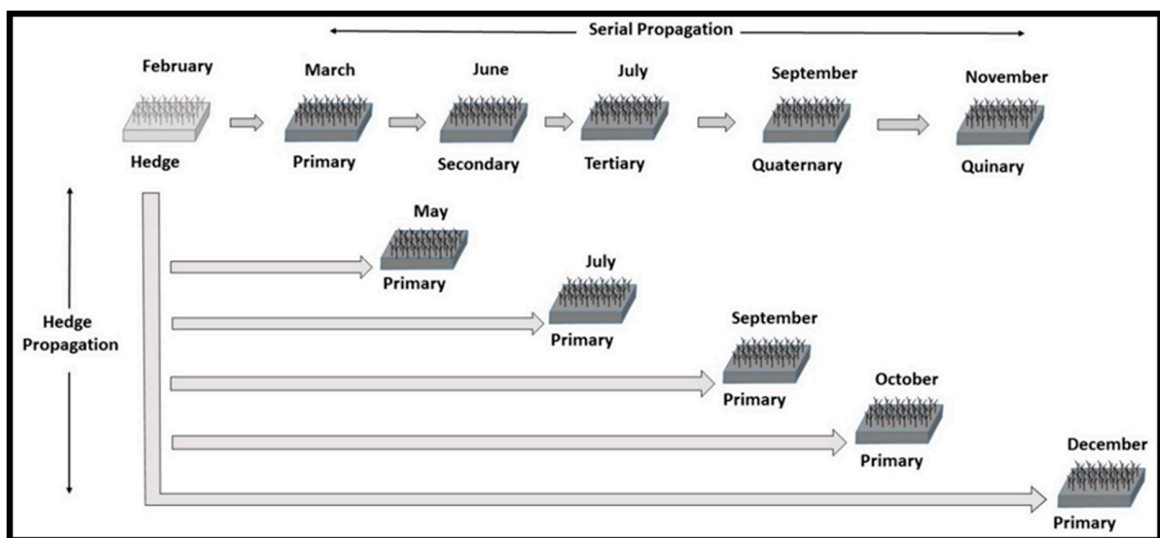

**Figure 1.** Schematic showing five cycles of greenhouse hedge and serial propagation.

**Table 2.** Greenhouse schedule of five season cycles of serial and hedge propagation.

| Propagation Cycle | Serial Propagation | | Hedge Propagation | |
|---|---|---|---|---|
| (Seasons) | Initiation | Harvest | Initiation | Harvest |
| 1 (Early spring–midsummer) | March 25 | June 2 | May 27 | July 18 |
| 2 (Late spring–summer) | June 2 | July 25 | July 18 | September 9 |
| 3 (Late summer–early autumn) | July 25 | September 19 | September 9 | October 21 |
| 4 (Autumn) | September 19 | November 9 | October 21 | December 12 |
| 5 (Late autumn–midwinter) | November 9 | December 30 | December 12 | February 13 |

### 2.3. Data Collection and Analysis

Survival was recorded as the percentage of ramets surviving each cycle of hedge or serial propagation; yield of macro-cuttings was specified in terms of the number of cuttings harvested per surviving ramet per cycle. Data were analyzed as a $2 \times 4$ factorial analysis of variance (i.e., two levels of propagation method, hedge versus serial, and four levels of hybrid variety (Equation 1). Both factors were considered fixed effects in the following analytical model:

$$Y_{ijk} = \alpha + \beta_i + \beta_j + \beta_{ij} + \varepsilon_{k(ij)}, \tag{1}$$

where $Y_{ijk}$ is ramet survival or macro-cutting yield for the $i$th variety using the $j$th propagation method during the $k$th propagation cycle, $\beta_i$ is the fixed effect of the $i$th variety, $\beta_j$ is the fixed effect of the $j$th method of propagation, $\beta_{ij}$ is the interaction of variety and propagation method, and $\varepsilon_{k(ij)}$ is experimental error. (The five cycles of propagation were nested within propagation method as replications since the timing of the serial and hedge propagation cycles were not coincident.)

Varietal and interaction effects were investigated using three single-degree of freedom a priori orthogonal contrasts based on the excessive and moderate levels of hardwood cutting production of the varieties in field nurseries (Table 1). These are: (1) Comparison of the *Populus* × *generosa* varieties (i.e., 790-99-28596 versus 893-01-31899,) (2) comparison of the *P. maximowiczii* interspecific varieties (i.e., 846-00-30120 versus 854-00-30517,) and (3) the varietal contrast between the two main taxonomic groups (i.e., contrast 1 versus contrast 2).

*2.4. Economic Analysis*

Survival and yield data from the greenhouse rooting trials were used in deriving the cost of producing containerized macro-cuttings. The internal rate of return on investment into improved planting stock produced by this system was then modeled as a function of increases in (1) yield and (2) pricing for feedstock that undergoes hydrolysis more cost-effectively. Research costs in breeding the improved varietals were not considered in the analysis; only the higher propagation and planting costs of containerized macro-cuttings were accounted for. (This approach assumes research and development costs are borne by a timber investment management organization that offers improved plant material exclusively to its investors at a price to cover propagation costs as part of its investment thesis.) Analyses were based on the average varietal propagation costs.

The analyses utilized a cost-of-production model developed by the USDA-NIFA AFRI CAP program, Advanced Hardwood Biofuels Northwest (AHB) for hybrid poplar as a biomass investment opportunity in renewable energy feedstock [18]. The model details the range of agronomic production activities and the costs of land leases, equipment, labor, chemicals and fuel needed to grow the crop and the expense of biomass harvesting and transportation. Inputs are integrated with yields and anticipated biomass pricing in deriving internal rates of return specific to poplar production regions in Washington, Oregon, Idaho, and Northern California, where four large-scale (i.e., 20–40 ha) hybrid poplar biomass AHB demonstration farms were managed from 2012 to 2018 [19]. The demonstration farms were situated on lands previously managed for grain, pasture, hay or row crops. The Oregon and Washington farms were unirrigated, while the Idaho and California farms were irrigated. Lease rates and production costs varied among farms, being highest in California due to the cost of land and irrigation and lowest in Washington where biomass cropping followed pasture (Table 3). The economic analyses supposed a medium-sized biorefinery supported by a 10,117 ha biomass farm managed for a 20-year rotation comprising a two-year establishment cycle followed by six three-year coppice cycles netting seven harvests. The planting density was set at 3,588 trees ha$^{-1}$. Plantation composition was set at five varieties each stocking 674 ha per each of the two years of planting before coppice regeneration was initiated. Based on the greenhouse propagation trials, the required number of rooted cuttings for the each of the first- and second-year's establishment was 2,424,035 per variety, after which, regeneration was transitioned to coppicing every three years.

**Table 3.** Land and biomass productions costs for Advanced Hardwood Biofuels Northwest (AHB) poplar demonstration farms throughout a 20-year rotation of seven harvest cycles.

| Farm Region | Lease Rate (USD ha$^{-1}$ year$^{-1}$) [1] | Production Cost (USD Dry Mg$^{-1}$ Rotation$^{-1}$) [2] |
|---|---|---|
| Washington | $111 | $77.70 |
| Oregon | $383 | $92.38 |
| Idaho | $175 | $99.46 |
| California | $865 | $146.28 |

[1] http://quickstats.nass.usda.gov/. [2] Biomass production costs are based on a yield of 16.5 Mg ha$^{-1}$ year$^{-1}$.

Internal rates of return were modeled across six levels of yield (9.0, 13.5, 18.0, 22.5, 27.0, 31.5 dry Mg ha$^{-1}$ year$^{-1}$ averaged over one 20-year rotation) and eight levels of market pricing ($66, $77, $88, $99, $110, $121, $132, and $145 dry Mg$^{-1}$ biomass) using land, management, and harvesting

costs specific to each AHB region (Table 3). The modeled yields up to 22.5 Mg ha$^{-1}$ year$^{-1}$ are consistent with AHB farm inventories and with growth rates projected for the AHB regions using the 3PG growth model (Physiological Processes Predicting Growth) [20]. Inclusion of yields of 27.0 and 31.5 Mg ha$^{-1}$ year$^{-1}$ in the analyses represents future genetic improvements. Likewise, modeled market prices ranging up to $110 Mg$^{-1}$ are consistent with supply simulations in the 2016 Billion Ton Study (BTS) and the Bioenergy KDF (Knowledge Discovery Framework) supplementary database [21]. This price range is in line with prices modeled for hardwoods managed as short rotation bioenergy crops elsewhere in the US [22,23]. Inclusion of market prices between $121 and $145 Mg$^{-1}$ in the analysis reflects future premiums for varietals capable of highly effective hydrolysis. Modeled results of investment into rapid multiplication were compared to a base case that assumed the standard method of supplying operational varieties in commercial quantities from nurseries as unrooted hardwood cuttings capable of yields of 16.5 Mg ha$^{-1}$ year$^{-1}$ based on prevailing yield estimates for the Pacific Northwest [2,24]. Prices modeled for the base case were not limited to the BTS range ($88–$110 Mg$^{-1}$), although it was realized that commercial poplar varieties have been developed for fiber and veneer markets, in which superior lignin chemistry that enables effective hydrolysis has not been an emphasis, and premium market pricing was not likely. The price of rooted planting stock, including packaging was gauged at approximately 4× the price of commercial hardwood cuttings in the base case. Likewise, the cost of cold storage for rooted cuttings, transportation of the rooted cuttings to the field, and the cost of planting the rooted cuttings was modeled at of 2×, 2×, and 6× of the respective cost for commercial hardwood cuttings.

## 3. Results

### 3.1. Laboratory

Micro-cuttings were successfully produced for all four varieties during in vitro propagation. The varieties were sub-cultured every two weeks during the shoot proliferation stage; between three and five in vitro cycles were required before enough high-quality micro-cuttings of each variety were available for the greenhouse rooting trials. Marked varietal differences were observed in the consistency of in vitro growth rates (Table 4). Variety 790-99-28596 was rated as a consistent grower, while varieties 854-00-30517 and 893-01-31899 exhibited sporadic growth. The slowest growth pattern was typical of variety 846-00-30120 that also showed the lowest multiplication rate. Multiplication rates were otherwise not too discrepant for the remaining three varieties, varying by just 5.3%.

**Table 4.** Varietal performance during in vitro propagation on two-week sub-culturing schedules.

| Variety | Taxon | Growth Pattern | Multiplication Factor [1] |
|---|---|---|---|
| 790-99-28596 | T × D | Consistent | 2.025 |
| 846-00-30120 | D × M | Slow | 1.680 |
| 854-00-30517 | T × M | Sporadic | 2.075 |
| 893-01-31899 | D × T | Sporadic | 2.133 |

[1] Number of quality micro-cuttings produced per transplant averaged over three-to-five sub-culture cycles following high grading.

### 3.2. Greenhouse

In vitro micro-cuttings were rooted nearly completely in the greenhouse for all varieties but one. Survival rates were recorded as 99.5% (790-99-28596), 87.3% (846-00-30120), and 100% (854-00-30517 and 893-01-31899). Considerable variation in greenhouse macro-cutting production was observed (Table 5). The number of macro-cuttings harvested during hedge propagation (2.992 cuttings plant$^{-1}$) exceeded the amount harvested during serial propagation (1.669 cuttings plant$^{-1}$) by 79% when averaged over the four varieties (Table 6). Likewise, substantial varietal variation in macro-cutting yield was

evidenced when averaged over the two propagation methods. Variety 854-00-30517 exhibited the highest mean macro-cutting yield (3.034 cuttings plant$^{-1}$), with varieties 790-99-28596 and 893-01-31899 showing the lowest yield, 1.959 and 1.754 cuttings plant$^{-1}$, respectively. Variety 846-00-30120 was intermediate in its production at 2.573 cuttings plant$^{-1}$ (Table 6). Correspondingly, the main effects of the variety and propagation method were both deemed highly significant sources of variation in macro-cutting yield in the analysis of variance (Table 5). The orthogonal contrasts suggested further that there were no significant yield differences for varieties within either the *P. × generosa* taxon or the *P. maximowiczii* interspecific taxa, but a large and significant effect associated with the third orthogonal contrast between the *P. × generosa* varieties (790-99-28596 and 893-01-31899) and the *P. maximowiczii* interspecific varieties (846-00-30120 and 854-00-30517).

**Table 5.** Analysis of variance of the yield of macro-cutting yield.

| Source of Variation | Degrees of Freedom | Sum of Squares | Mean Square | F Ratio |
|---|---|---|---|---|
| Variety | 3 | 10.2443 | 3.4148 | 18.52 * |
| Within *P. × generosa* | 1 | 0.2110 | 0.2110 | 1.1444 |
| Within *P. maximowiczii* hybrids | 1 | 1.0635 | 1.0635 | 5.7673 |
| Between *P. × generosa* and *P. maximowiczii* | 1 | 8.9698 | 8.9698 | 48.6429 * |
| Propagation Method | 1 | 17.5001 | 17.5001 | 94.89 * |
| Variety-by-Propagation Method | 3 | 8.7568 | 2.9189 | 15.83 * |
| Within *P. × generosa* | 1 | 0.0127 | 0.0127 | 0.0688 |
| Within *P. maximowiczii* hybrids | 1 | 0.0007 | 0.0007 | 0.0038 |
| Between *P. × generosa* and *P. maximowiczii* | 1 | 8.7434 | 8.7434 | 47.4156 * |
| Error | 32 | 5.9018 | 0.1844 | |
| Total | 39 | 42.4030 | | |

\* Significant at the 0.0001 probability level.

**Table 6.** Macro-cutting survival and yield (cuttings plant$^{-1}$) during greenhouse propagation [1].

| Variety | Taxon | Serial Propagation | | Hedge Propagation | | Mean | |
|---|---|---|---|---|---|---|---|
| | | Survival (%) | Yield | Survival (%) | Yield | Survival (%) | Yield |
| 790-99-28596 | T × D | 98.1 | 1.791 | 95.9 | 2.128 | 97.1 | 1.959 |
| 846-00-30120 | D × M | 97.5 | 1.438 | 97.4 | 3.708 | 95.9 | 2.573 |
| 854-00-30517 | T × M | 94.1 | 1.911 | 96.2 | 4.157 | 97.9 | 3.034 |
| 893-01-31899 | D × T | 95.0 | 1.535 | 95.1 | 1.973 | 94.9 | 1.754 |

[1] Standard errors of varietal, propagation, and interaction means for yield are 0.134, 0.096, and 0.192 cuttings plant$^{-1}$, respectively. Standard errors of survival means are 1.03, 0.73, and 1.46%, respectively.

Very little varietal variation was recorded in survival during the macro-cutting propagation trials, averaging 96.6% for both serial and hedge propagation, with a standard deviation among varieties of 1.92% (serial propagation) and 1.23% (hedge propagation) (Table 6). Consequently, analysis of variance did not show any significant main or interaction effects of survival (Supplementary Table S1).

The first order interaction of variety and propagation method was also highly significant (Table 5). Orthogonal contrast revealed that the interaction again resulted mainly from the much stronger response of the two *P. maximowiczii* hybrids to hedging in comparison to the *P. × generosa* varieties (Figure 2).

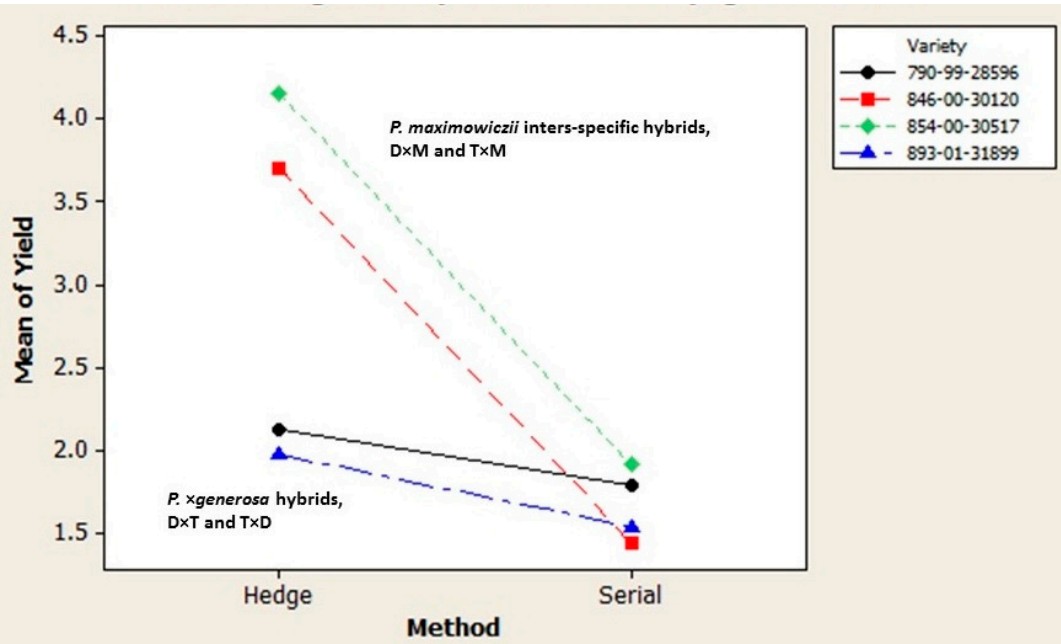

**Figure 2.** Varietal variation in the response of macro-cutting yield to propagation method.

### 3.3. Economic Analysis

Using the survival and yield data from the greenhouse rooting trials averaged over varieties (Table 6), the production of the 2.4 million rooted cuttings was projected in terms of five cycles of hedge and serial macro-cutting propagation beginning in the spring and concluding in the winter following the delivery of the micro-cuttings. To produce this quantity of macro-cuttings, it was estimated that 29,000 in vitro micro-cuttings were needed based on the mean varietal yield of macro-cuttings during greenhouse propagation.

Investment returns under the rapid multiplication scenario for each of the four demonstration regions are detailed in Table 7 as a function of biomass yield and market pricing. The profile of internal rates of return (IRRs) at the two non-irrigated farms in Oregon and Washington was superior to the irrigated Idaho and California farms. The low least rate at the Idaho farm partially offset the cost of irrigation, with better returns compared to California. The percentage of positive IRRs recorded for the 48 scenarios of yield and market prices at each demonstration farm was 75% for Washington (36 positive returns), 58% for Oregon (28 positive returns), 50% for Idaho (24 positive returns), and 25% (12 positive returns) for California. Averaged over yields and market prices, the mean for the positive IRRs for Washington, Oregon, Idaho, and California was 15.91%, 13.89%, 13.35%, and 6.89%, respectively. The maximum return in Washington 37.90% and 31.96% in Oregon. Maximum returns for the other two regions were 29.42% (Idaho) and 16.73% (California). Returns estimated for Washington were the most variable, with a standard deviation of 10.30%, followed by Oregon (8.60%) Idaho (7.71%), and California (5.35%).

Both the mean and the number of positive IRRs (*n*) for the rapid multiplication scenario increased steadily within the boundaries of the BTS market prices ($66–$110 Mg$^{-1}$) when results were averaged by yield and compiled for all farms. The progression was 3.29% (*n* = 2) at $66 Mg$^{-1}$, 5.76% (*n* = 6) at $77 Mg$^{-1}$, 9.02% (*n* = 9) at $88 Mg$^{-1}$, 10.14% (*n* = 13) at $99 Mg$^{-1}$, and 12.63% (*n* = 15) at $110 Mg$^{-1}$. Positive values of IRR recorded above the BTS price range ($121–145 Mg$^{-1}$) were 15.37% (*n* = 16) at $121 Mg$^{-1}$, 16.01% (*n* = 19) at $132 Mg$^{-1}$, and 18.55% (*n* = 20) at $145 Mg$^{-1}$. Likewise, the number of positive IRRs and their mean value increased with yield when averaged across market prices being 2.09% (*n* = 2), 6.10% (*n* = 10), 9.87% (*n* = 16), 13.41% (*n* = 20), 15.23% (*n* = 25), and 18.23% (*n* = 27), respectively, for the 9.0, 13.5, 18.0, 22.5, 27.0, and 31.5 Mg ha$^{-1}$ year$^{-1}$ yield levels. These IRRs compare

to an upper bound of IRRs for hybrid poplar bioenergy programs in the northeastern US of 14.6% at prevailing stumpage prices of \$11 Mg$^{-1}$ and not accounting for harvest and transportation costs [25].

　　Returns from investment into rapid multiplication frequently surpassed those of the base case analysis within the range of the BTS prices across the four farm regions. Base case analyses (yield set to 16.5 Mg ha$^{-1}$ year$^{-1}$) showed no positive IRRs at the \$66 and \$77 Mg$^{-1}$ pricing levels at any farm, and only one positive return was recorded at the \$88 Mg$^{-1}$ price level (8.81%, Washington) (Table 7). Two positive IRRs were recorded for the base case at the \$99 market price, averaging 9.87% at Washington and Oregon; these increased to a mean IRR of 12.31% at the \$110 price level averaged across Washington, Oregon, and Idaho. The base case IRRs at the \$99 and \$110 Mg$^{-1}$ price points exceeded those observed for rapid multiplication at Oregon and Idaho, when newly released varieties produced no more than 18.0 Mg ha$^{-1}$ year$^{-1}$. The base case analysis for Washington and Oregon was marginally better than the rapid multiplication analysis at yields of 22.5 Mg ha$^{-1}$ year$^{-1}$ across a price range of \$110–\$145 Mg$^{-1}$ (Table 7). However, when the comparison was made for new varieties characterized by yields at the two uppermost levels (27.0 and 31.5 Mg ha$^{-1}$ year$^{-1}$,) the rapid multiplication scenario was uniformly superior to the base case analysis both within and above the price BTS range at Washington, Oregon, and Idaho farms. No positive IRRs were returned for the base case analysis at California.

**Table 7.** Internal rates of return (%) as a function of yield and market pricing [1].

| Yield (Mg ha$^{-1}$ year$^{-1}$) | Price (USD Mg$^{-1}$) | | | | | | | | (+) IRRs | |
|---|---|---|---|---|---|---|---|---|---|---|
| **Washington** | \$66 | \$77 | \$88 | \$99 | \$110 | \$121 | \$132 | \$145 | Mean | N |
| 9.0 | | | | | | | 0.54 | 3.63 | 2.09 | 2 |
| 13.5 | | | | 0.57 | 4.48 | 7.56 | 10.19 | 12.98 | 7.16 | 5 |
| 18.0 | | | 3.57 | 8.05 | 11.64 | 14.76 | 17.57 | 20.68 | 12.71 | 6 |
| 22.5 | | 3.84 | 9.41 | 13.70 | 17.37 | 20.67 | 23.71 | 27.11 | 16.54 | 7 |
| 27.0 | 0.91 | 8.78 | 14.1 | 18.50 | 22.33 | 25.85 | 29.12 | 32.81 | 19.05 | 8 |
| 31.5 | 5.66 | 12.80 | 18.20 | 22.70 | 26.74 | 30.47 | 33.95 | 37.90 | 23.55 | 8 |
| Base case (16.5) | | | 8.81 | 14.9 | 19.96 | 24.42 | 28.51 | 33.09 | 21.62 | 6 |
| **Oregon** | \$66 | \$77 | \$88 | \$99 | \$110 | \$121 | \$132 | \$145 | Mean | N |
| 9.0 | | | | | | | | | 0.00 | 0 |
| 13.5 | | | | | | 2.31 | 5.44 | 8.57 | 5.44 | 3 |
| 18.0 | | | | 1.95 | 6.26 | 9.74 | 12.76 | 16.00 | 9.34 | 5 |
| 22.5 | | | 2.83 | 7.89 | 11.92 | 15.41 | 18.56 | 22.03 | 13.11 | 6 |
| 27.0 | | 1.15 | 7.64 | 12.50 | 16.57 | 20.21 | 23.55 | 27.28 | 15.55 | 7 |
| 31.5 | | 5.41 | 11.5 | 16.40 | 20.60 | 24.43 | 27.98 | 31.96 | 19.75 | 7 |
| Base case (16.5) | | | | 4.80 | 10.69 | 15.45 | 19.63 | 24.17 | 14.95 | 5 |
| **Idaho** | \$66 | \$77 | \$88 | \$99 | \$110 | \$121 | \$132 | \$145 | Mean | N |
| 9.0 | . | | | | | | | | 0.00 | 0 |
| 13.5 | | | | | | | 2.79 | 6.12 | 4.46 | 2 |
| 18.0 | | | | | 3.54 | 7.22 | 10.33 | 13.61 | 8.68 | 4 |
| 22.5 | | | | 5.25 | 9.45 | 13.00 | 16.16 | 19.61 | 12.69 | 5 |
| 27.0 | | | 4.96 | 10.00 | 14.14 | 17.79 | 21.11 | 24.8 | 15.47 | 6 |
| 31.5 | | 2.53 | 8.99 | 13.94 | 18.17 | 21.98 | 25.49 | 29.42 | 17.22 | 7 |
| Base case (16.5) | | | | | 6.27 | 11.21 | 15.40 | 19.88 | 13.19 | 4 |
| **California** | \$66 | \$77 | \$88 | \$99 | \$110 | \$121 | \$132 | \$145 | Mean | N |
| 9.0 | | | | | | | | | 0.00 | 0 |
| 13.5 | | | | | | | | | 0.00 | 0 |
| 18.0 | | | | | | | | 0.28 | 0.28 | 1 |
| 22.5 | | | | | | | 3.28 | 7.11 | 5.20 | 2 |
| 27.0 | | | | | 0.76 | 5.08 | 8.64 | 12.33 | 6.70 | 4 |
| 31.5 | | | | 0.44 | 5.47 | 9.51 | 13.00 | 16.73 | 9.03 | 5 |
| Base case (16.5) | | | | | | | | | 0.00 | 0 |

[1] Empty cells signify negative internal rates of return.

## 4. Discussion

The present study looked at returns that accrue over a 20-year rotation when rapid multiplication is used to produce planting stock of new varieties directly for operations, compared with the coincident use of less expensive hardwood cuttings to initiate planting with standard varieties that are available in commercial quantities. Thus, this comparison focused on the last component of the entire tree improvement cycle when newly-released varieties are propagated en masse as containerized rooted cuttings to originate a bioenergy planting that later transitions to serial coppice regeneration. An implicit assumption is that if the route of rapid propagation of new varieties is chosen, that process takes place at the same time that land is acquired and prepared, meaning that the new varieties are available at the corresponding time that standard varieties are purchased from a commercial nursery. Returns were estimated under both options over the course of a 20-year rotation. Research costs in developing new hybrids were not factored into the analysis, reasoning that a timber investment management organization produces proprietary varieties as leverage in raising capital into its bioenergy investment funds.

Investments into hybrid poplar bioenergy operations at all four AHB regions with current commercial varieties were previously shown to generate insufficient returns to attract private-sector capital and indicated the need for marked increases in biomass pricing and yields [19]. Other AHB analyses have suggested that the adoption of poplar feedstock operations is also impacted by the opportunity cost and the demand for competing crop commodities [26]. Despite the influence of biomass yields, selling prices and alternative crop options, dedicated energy crops are nonetheless expected to provide a significant component of biorefinery feedstock supply in the AHB region at a benchmark biofuel price of \$19.6 GJ$^{-1}$ [27].

Rapid multiplication was clearly superior to the base case at the two highest rates of yield irrespective of market pricing and region. For investments into rapid multiplication to make sense at the Washington, Oregon, and Idaho locations within the BTS range, yields of 22.5 Mg ha$^{-1}$ year$^{-1}$ and above would be needed for the most part. Conversely, a decision to forego investment into rapid multiplication in favor of the base case analysis of current commercial varieties would be generally expected at yields of 18–22.5 Mg$^{-1}$ year$^{-1}$ at all but the California location; this is true especially at market prices above the BTS range. However, it is doubtful whether current commercial hybrid poplar varieties that have been bred and selected for markets other than bioenergy would have the preferred biomass chemistry that would incentivize the market to offer a premium represented by prices above the BTS range. The regional influence of the study was noteworthy: The most favorable investment analysis, in terms of the overall mean and number of positive IRRs, was noted for the two non-irrigated farms, Washington and Oregon; of the two, the former is pasture land with the lowest lease rate in the study, while the Oregon farm was situated on cropland that commanded a higher lease rate. The two dryland farms requiring irrigation-Idaho and California-had lower overall IRRs and counts of positive returns. The California farm with the highest lease rate in the study exhibited the least favorable profile of returns under both the rapid multiplication and the base case. Returns at California under the base case were uniformly negative, and appreciable returns for the most part under the rapid multiplication case were not seen until yields of 27 and 31.5 Mg$^{-1}$ year$^{-1}$ at market prices above the BTS range.

The argument for rapid multiplication is that the exploitation of gains in biomass yield and chemistry requires a swift and seamless introduction of those gains into operations that regenerate by frequent coppicing to justify the expense of propagation [7]. However, are the genetic gains in yield and feedstock quality presented in this study realistic as requisites to poplar bioenergy farms investment? They may be. Poplar growth and yield simulations support the yields modeled in this study up to 22.5 to 27.0 Mg ha$^{-1}$ year$^{-1}$ [25,28,29], and preliminary estimates from bioenergy plantation density trials of 67 hybrid poplar varieties support theoretical yields of 35 Mg ha$^{-1}$ year$^{-1}$ for the most productive varieties [30]. Achieving yields above this level through ongoing hybridization is conceivable based on projections of 90% biomass gains for hybrid poplar when using reciprocal, intraspecific recurrent breeding in advance of first generation interspecific hybridization and within-family clonal selection [31].

Similarly, to accomplish improvements in biomass quality, gains in cell wall deconstruction, sugar release and biomass hydrolyzability may be anticipated using genomic selection of naturally-occurring mutant alleles in the lignin biosynthetic pathway [32–34], as well as lignin transformation [35,36], with the expectation that low-lignin biomass may significantly reduce the operating cost of mechanical and chemical pre-treatment. To illustrate, research at Oak Ridge National Laboratory's Bioenergy Research Center demonstrated a 15% increase in sugar yield associated with the biochemical conversion of low-lignin poplar varieties [37]. AHB research confirms this with reports of increases of 19% in sugar yield leading to 10% increases in biorefinery revenue that accompany individual hybrid selections [38]. However, it should be recognized that achieving these gains in large-scale plantations where soil quality and water supply vary at the landscape scale will be challenging.

Although gains in both biomass quantity and quality are likely, genetic improvement in the former is preferred from the perspective of an investor into biomass farms: Farm management is far better positioned to exploit increases in yield rather than bargaining the price at which the biomass is sold in to the energy markets. Increased yield directly provides incremental revenue, while an increase in price could be difficult to negotiate to the investor's favor, recognizing that biorefineries may not be able to adjust their processes to specific biomass characteristic-the superiority of cell wall chemistry notwithstanding-when managing a diversity of cellulosic feedstock sources.

Beyond the genetic gains that strengthen the attractiveness of rapid multiplication, its appeal may be improved per se by exploiting differences in the varietal performance during both the in vitro and greenhouse propagation phases [39–41]. For instance, the statistically-significant yield differences in macro-cutting propagation noted can be exploited to optimize the system, as recognized elsewhere [42]. This is especially true of the *P. maximowiczii* interspecific varieties and their strong response to hedge propagation that increased cutting yield by approximately twofold. The superior greenhouse performance of the *P. maximowiczii* hybrids mirrored their exceptional height growth and extremely low rates of sylleptic branching that maximizes nursery production of hardwood cuttings. Beyond these manipulations, the process can be shortened considerably using test trees from final stage yield trials to collect succulent macro-cuttings during spring shoot extension which are then used to initiate greenhouse propagation circumventing the in vitro production of micro-cuttings with cost and time savings.

Finally, there are other valuable features of the rapid multiplication system associated with the use of containerized rooted planting stock that reduce the risk of plantation failures. First, vagaries of the weather-frost injury and drought-are much more easily mitigated when planting stock is grown in controlled greenhouse environments in comparison to field nurseries. Second, greenhouse production of newly-deployed varieties allows for far greater flexibility than field nurseries when turning over and initiating new varieties in to scale-up operations when, for example, such varieties must be removed from production due to a loss of pest resistance or a change in market preference. Third, rooted cuttings greatly reduce the risk of planting failures compared to the risk incurred using unrooted hardwood cuttings when an unfavorable environment-protracted dry or cold periods-is encountered during the time of establishment.

## 5. Conclusions

1. No variety failed to respond to in vitro micro-cutting propagation or greenhouse macro-cutting propagation.

2. The *Populus deltoides* × *P. maximowiczii* and the *Populus trichocarpa* × *P. maximowiczii* varieties greatly exceeded the multiplication rate of the *P.* × *generosa* varieties under greenhouse propagation. This was largely due to their exceptional response to repeated hedging required to initiate multiple tracks of serial propagation. It mirrored the stronger performance of the *P. maximowiczii* taxon under traditional stoolbed culture.

3. Rapid multiplication was clearly superior to the base case at the two highest rates of yield irrespective of market pricing and region. For investments into rapid multiplication to make sense at

the Washington, Oregon, and Idaho locations within the BTS range, yields of 22.5 Mg ha$^{-1}$ year$^{-1}$ and above would be needed for the most part.

4. Conversely, a decision to forego investment into rapid multiplication in favor of the base case analysis of current commercial varieties would be generally expected at yields of 18.0–22.5 Mg$^{-1}$ year$^{-1}$ at all but the California location; this is true especially at market prices above the BTS range. However, it is doubtful whether current commercial hybrid poplar varieties for which preferred biomass chemistry has not been an improvement emphasis would justify market prices above the BTS range.

**Supplementary Materials:** The following are available online at http://www.mdpi.com/1999-4907/10/5/446/s1, Table S1: Analysis of variance in survival.

**Author Contributions:** Conceptualization, B.J.S. and C.G.; Data curation, R.A.S.; Investigation, B.J.S.; Methodology, K.H., C.G. and J.E.; Project administration, B.J.S.; Resources, J.E.; Supervision, B.J.S. and R.A.S.; Validation, R.A.S.; Writing—original draft, B.J.S.; Writing—review and editing, R.A.S.

**Funding:** This research was funded by the United States Department of Agriculture, National Institute for Food and Agriculture, Grant no. 2011-68005-30407, Agriculture and Food Research Initiative.

**Acknowledgments:** The authors gratefully acknowledge the support of the United States Department of Agriculture National Institute of Food and Agriculture (USDA-NIFA). The work reported here was conducted under Agriculture and Food Research Initiative (AFRI) Competitive Grant Number 2011-68005-30407.

**Conflicts of Interest:** The authors declare no conflict of interest. The funders had no role in the design of the study; in the collection, analyses, or interpretation of data; in the writing of the manuscript, or in the decision to publish the results.

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
