# Peer review of "The Economics of Rapid Multiplication of Hybrid Poplar Biomass Varieties"

_forests, doi:10.3390/f10050446_

Reviewer 1 Report

Review report  

The Economics of Rapid Multiplication of Hybrid Poplar Biomass Varieties

 General comments

This study address a very important topic for the large-scale deployment of woody biomass production with improved poplar clones. I have a few recommendations that would contribute to improving the clarity of the study. Some aspects of data presentation and statistics should also be improved.

Specific Comments

L77. Edge propagation. I think the authors should clearly define what is edge propagation? This propagation technique is poorly documented on the web and it may be unfamiliar to readers who are not specialist in the field of poplar propagation. Also, I strongly encourage the authors to add a figure with photographs showing in vitro micro-cuttings, and hedge and serial propagation technique.  

Table 5, Fig. 2-3. A measure of variability (SE, SD or CI) should be included in the table of figures, not stated in the text. P-value of the tested effects with ANOVA should also be included in the table or in the table or figure legends. The tiles embedded in Fig.2-3 should also be removed, as a figure caption is already provided.  

Table 6. I think this table should be placed before Table 5. First, ANOVA output results are presented, then results for the significant interaction effect on yield are presented. Also, the use of contrasts requires a priori hypothesis (set in advance, before the ANOVA is run). These hypotheses should appear somewhere in the introduction and be supported by anterior studies or observations.   

Fig. 3. I encourage the authors to run separate ANOVA (one per propagation method) to test the seasonal effect on yield. The author claim that season had a strong effect on macro-cutting yield (L29-30 of the abstract). However, the significance of this seasonal effect was not statistically tested.   

Some Results in Table 5 and Figure 2 are the same. I think it’s better to keep Fig. 2, which more clearly show the interaction effect. ANOVA Results (p-value and treatment means) for survival could be put in the text of the Materials and Methods section.  I don’t think that small SD (L142) is a good argument to not run an ANOVA for the survival data.  

L255-256. «The significance of the varietal effect was nearly all due to the third orthogonal contrast». ANOVA and contrasts are two different analyses. Therefore, it cannot be claimed that a significant effect tested with ANOVA is because of a contrast. ANOVA test effects and contrasts test specific hypotheses between particular sets of means. Rather, it could be stated that the ANOVA detected a significant varietal effect (p-value) and a significant variety x propagation method interaction effect (p-value). Contrast analysis further suggest that there were no significant yield differences for clones within P. generosa clones and P. maximowiczii hybrids, but large and significant (p-value) differences between P. generosa clones and P. maximowiczii hybrids.   Table 7. I would be interesting to add IRR for a yield of 16.5 t/ha/yr with new propagation technique vs. base case scenario (yield of 16.5 t/ha/yr with hardwood cuttings). This would give a clearer perspective as IRR for both propagation techniques would be compared with the same yield.

L303-311. Those data should appear in Table 7 (mean IRR and number of positive IRR). This would greatly clarify data presentation.

L374-376. Expected yields are quite high (up to 35 t/ha/yr). Are these yields realistic for large-scale plantations where soil quality and water supply will surely vary at the landscape scale. This should be more fully discussed as research plots are generally small compared to operational plantations.

L401-403. Strange parentheses.    

Author Response

L77. Edge propagation. I think the authors should clearly define what is edge propagation? This propagation technique is poorly documented on the web and it may be unfamiliar to readers who are not specialist in the field of poplar propagation. Also, I strongly encourage the authors to add a figure with photographs showing in vitro micro-cuttings, and hedge and serial propagation technique.  

The definition of in vitro, hedge and serial propagation has been added (line 81) in the vicinity of the introduction where micro- and macro-cuttings are also defined.  A new citation on which the definitions are based is included in the revised manuscript (Ahuja, M. R.; Libby, W. J.  Glossary. In Clonal Forestry I Genetics and Biotechnology.  Ahuja, M. R.; Libby, W. J., Eds.; Springer-Verlag. Berlin Heidelberg, 1993; Germany. Pp 255-265.)  Unfortunately, I have no good quality image showing the hedge propagation system.

Table 5, Fig. 2-3. A measure of variability (SE, SD or CI) should be included in the table of figures, not stated in the text. P-value of the tested effects with ANOVA should also be included in the table or in the table or figure legends. The tiles embedded in Fig.2-3 should also be removed, as a figure caption is already provided.  

Standard errors for main effects and interaction means have been added to table 5. (Now table 6.)

The P-value of all tested effects in the analysis of variance table (table 6 previously, now changed to table 5) is included in a footnote to the table as all effects tested to the same P-value.

The embedded title in Figure 2 has been removed.  (Figure 3 has been removed from the draft.  See comment 4 below.)

Table 6. I think this table should be placed before Table 5. First, ANOVA output results are presented, then results for the significant interaction effect on yield are presented. Also, the use of contrasts requires a priori hypothesis (set in advance, before the ANOVA is run). These hypotheses should appear somewhere in the introduction and be supported by anterior studies or observations.   

The order of tables 5 and 6 have switched.   Table numbers and references to the reordered tables have been changed in the manuscript to coincide with the revised numbering of the tables.

This is an excellent and erudite comment of the reviewer. The construction of the orthogonal contrasts was based on a priori hypothesis formed on the performance of the four varieties in commercial propagation nurseries as presented in table 1.  This logic is now described in the Methods section beginning at line 165. 

Fig. 3. I encourage the authors to run separate ANOVA (one per propagation method) to test the seasonal effect on yield. The author claim that season had a strong effect on macro-cutting yield (L29-30 of the abstract). However, the significance of this seasonal effect was not statistically tested.   

  In the Methods section, Equation 1 (line 159) shows that the propagation season (symbolized as the kth component in the equation) was not considered as an experimental treatment in the factorial study design.  Rather, propagation season served as replication to estimate the variability associated with the two experimental factors (variety and propagation method.) As such, propagation season was listed as the error term of Equation 1 (symbolized εk(ij),) thus allowing the effects of variety and propagation method to be considered in light of their replication across the five seasons.  I am hesitant in redefining season as a treatment effect in a new model and testing its significance.  I have fully considered the importance of the review comment, but I think it best to take this conservative position to the study’s statistical analysis. To do otherwise seems to be extracting more information than what the original experimental design supports.  Accordingly, figure 3 and the pertinent passages in the Abstract, Results and Discussion sections referencing the seasonal effect of the propagation system have been removed from the manuscript.

Some Results in Table 5 and Figure 2 are the same. I think it’s better to keep Fig. 2, which more clearly show the interaction effect. ANOVA Results (p-value and treatment means) for survival could be put in the text of the Materials and Methods section.  I don’t think that small SD (L142) is a good argument to not run an ANOVA for the survival data.   

An analysis of variance has been conducted for survival and included in the submission as supplementary table 1.

Figure 2 has been improved and retained in the text.

L255-256. «The significance of the varietal effect was nearly all due to the third orthogonal contrast». ANOVA and contrasts are two different analyses. Therefore, it cannot be claimed that a significant effect tested with ANOVA is because of a contrast. ANOVA test effects and contrasts test specific hypotheses between particular sets of means. Rather, it could be stated that the ANOVA detected a significant varietal effect (p-value) and a significant variety x propagation method interaction effect (p-value). Contrast analysis further suggest that there were no significant yield differences for clones within P. generosa clones and P. maximowiczii hybrids, but large and significant (p-value) differences between P. generosa clones and P. maximowiczii hybrids.  

The point is well taken.  The relationship between ANOVA results and the orthogonal contrasts have been clarified (lines 257-263) drawing on the reviewer’s text above.

Table 7. I would be interesting to add IRR for a yield of 16.5 t/ha/yr with new propagation technique vs. base case scenario (yield of 16.5 t/ha/yr with hardwood cuttings). This would give a clearer perspective as IRR for both propagation techniques would be compared with the same yield.

I prefer not to include this comparison for the following reason.  The intent of the study was to compare the return on investment into rapid propagation in accelerating the scale-up of new varieties that had been bred for improved yield or improved biomass chemistry or both.  The base of comparison was the use of pre-existing standard varieties (represented by a yield of 16 Mg ha-1 yr-1) that are immediately available in commercial quantities as hardwood cuttings.  The investment question that this study posed was, at what level of genetic gain(s) is it worthwhile to invest in expensive rapid propagation when the only other option is to purchase hardwood cuttings of standard varieties.   The study did not consider the option of rapid propagation of standard varieties since they are readily available in the nursery trade.  Basically, the return on investment into rapid scale-up of better varieties was weighed against the opportunity cost of buying hardwood cuttings of standard clones from nurseries selling material that had been already multiplied to scale. The considerable expense of rapid propagation for a one-time planting before coppice regeneration begins in dedicated biomass energy plantations is only justified by superior plant material.  There is, perhaps a nuanced distinction here, but inclusion of the rapid propagation of standard varieties that are already available from nurseries would, I fear, compromise the study’s intent.     

L303-311. Those data should appear in Table 7 (mean IRR and number of positive IRR). This would greatly clarify data presentation.

Agreed.  The mean and number of positive IRRs have been added to table 7.

L374-376. Expected yields are quite high (up to 35 t/ha/yr). Are these yields realistic for large-scale plantations where soil quality and water supply will surely vary at the landscape scale. This should be more fully discussed as research plots are generally small compared to operational plantations.

A good point.  A statement has been added (lines 422-423) acknowledging the challenge of achieving the upper bounds of the modeled gains in yield and quality under operational conditions.

L401-403. Strange parentheses.

The parentheses have been removed.

Reviewer 2 Report

The thesis is focused on interdisciplinary topic including the both dendrology experiment on Poplar clones and economy evaluation of profitability. The aim of the manuscript was to assess the impact of gains in biomass yield and quality on investment in rapid propagation techniques that speed to commercial deployment. Basic data were obtained on greenhouse experiments, a new characteristic of the internal rate of return and adequate statistical processing were used. The results show promising poplar clones, including the propagation method.

I have no scientific advice or comments, only two formal recommendations. I personally appreciate the careful processing of the whole text, especially the discussion.

My short formal recommendation:
Table 6: align numbers in columns
Abstract: each subchapter (Background, Objective, Material and Methods, ...) should start on a new line.

Author Response

The abstract and table 6 (now table 5) have been re-formatted.